# The Effectiveness of the Association of Chlorhexidine with Mechanical Treatment of Peri-Implant Mucositis

**DOI:** 10.3390/healthcare11131918

**Published:** 2023-07-03

**Authors:** Anca Silvia Dumitriu, Stana Păunică, Ximena Anca Nicolae, Dana Cristina Bodnar, Ștefan Dimitrie Albu, Ioana Suciu, Dragoș Nicolae Ciongaru, Marina Cristina Giurgiu

**Affiliations:** 1Department of Periodontology, Faculty of Dental Medicine, University of Medicine and Pharmacy “Carol Davila”, 37 Dionisie Lupu Street, 020021 Bucharest, Romania; anca.dumitriu@umfcd.ro (A.S.D.); stefan-dimitrie.albu@drd.umfcd.ro (Ș.D.A.); nicolae-dragos.ciongaru@drd.umfcd.ro (D.N.C.); marina.giurgiu@umfcd.ro (M.C.G.); 2Doctoral School, Faculty of Dental Medicine, University of Medicine and Pharmacy “Carol Davila”, 37 Dionisie Lupu Street, 020021 Bucharest, Romania; ximena-anca.nicolae@drd.umfcd.ro; 3Department of Restorative Odontotherapy, Faculty of Dental Medicine, University of Medicine and Pharmacy “Carol Davila”, 37 Dionisie Lupu Street, 020021 Bucharest, Romania; 4Department of Endodontics, Faculty of Dental Medicine, University of Medicine and Pharmacy “Carol Davila”, 37 Dionisie Lupu Street, 020021 Bucharest, Romania; ioana.suciu@umfcd.ro

**Keywords:** chlorhexidine, peri-implant mucositis, oral hygiene, non-surgical treatment

## Abstract

(1) Background: The aim of the study was to evaluate the benefit of combining chlorhexidine with the mechanical treatment of peri-implant mucositis. (2) Methods: Articles from 2016 to 2021 included in the PubMed and Scopus databases were analyzed, following the PICOS criteria and the randomized controlled study model that used chlorhexidine in various forms in the treatment of peri-mucositis. According to the established criteria, a limited number of studies were selected. These studies had as their criteria of evaluation for the effectiveness of chlorhexidine, plaque indices, bleeding indices and depth probing indices. Chlorhexidine has been used after mechanical debridement as a solution, with different concentrations of 0.06%/0.12%/0.2% alone or in a concentration of 0.03%, in combination with 0.05% cetylpyridinium chloride, as well as in the form of a gel with a concentration of 0.2%. (3) Results: The results were assessed to a placebo or other substances, and showed a significant reduction in the indices with a follow-up period ranging from 3 months to 1 year. (4) Conclusions: The association of chlorhexidine with the mechanical treatment of peri-implant mucositis has a role in reducing inflammation, although a complete remission was not obtained in all cases, and the results were not statistically significantly different from the use of other antiseptics.

## 1. Introduction

Over the past decade, the field of implant–prosthetic therapy for edentulous patients has made remarkable progress. This advancement has not only addressed the issue of occlusal rehabilitation, but has also highlighted concerns regarding the oral mucosa pathology surrounding the implants. Peri-implant mucositis is a chronic reversible inflammation of the soft tissues around the implant and if left untreated, it can progress to the destruction of bone support with the formation of peri-implantitis, compromising the stability of the implant [1,2]. The most suggestive clinical sign for the diagnosis of peri-implant mucositis is bleeding on probing [3] and, according to the Classification of Periodontal and Peri-Implant Diseases and Conditions (2018), is diagnosed as inflammation of the peri-implant mucosa (bleeding on probing, erythema, swelling and suppuration) and the absence of continuing marginal peri-implant bone loss [1]. The development of peri-implant mucositis, similar to periodontal disease, is primarily attributed to bacterial overgrowth, particularly within the biofilm located at the submucosal region near the implant–prosthetic abutment interface [4]. Thus, it is mandatory to elaborate a treatment plan for the efficient removal of bacterial plaque, in order to reduce the inflammatory signs [5]. The treatment of peri-implant mucositis involves a combination of debridement operations of the supra- and submucosal surfaces of the implant, associated with the administration of various antiseptic and antibiotic substances used locally and/or systemically depending on the particularities of each case, along with instructions for efficient oral care performed by the patient [3,6,7,8,9].

Currently, the accepted notion is that employing an appropriate brushing technique is adequate to maintain plaque and gingival bleeding indices within normal ranges, even without the use of anti-plaque mouthwashes [10,11,12,13]. However, most studies have shown that the use of mouthwash with chlorhexidine (CHX) is necessary for the therapy of periodontal patients, as it is the most-used antiseptic after surgical periodontal therapy [14,15]. Chlorhexidine digluconate (C_34_H_54_Cl_2_N_10_O_14_) is a broad-spectrum antiseptic substance, from a class of biguanides with topical action on bacteria, fungi and viruses [14]. It has a cationic compound that exerts its actions on immediate contact with microorganisms whose surfaces are negatively polarized, targeting the cell wall [14]. Thus, this explains its higher efficiency on Gram-positive bacteria, which have a higher negative charge compared to Gram-negative bacteria [14,16]. On the surface of the teeth, CHX adheres to the salivary proteins and exfoliated epithelial cells, blocks the acidic valences of the glycoproteins in saliva and decreases glycoprotein assimilation, preventing pellicle formation [17,18]. CHX has an extended antiseptic release, blocks the formation of bacterial plaque and competes with calcium ions, preventing the binding of mature plaque [19,20,21]. CHX is used in periodontal treatment in various forms: solution (with concentrations range between 0.03–0.2%), sprays, toothpaste, gels (1%), varnishes, periodontal chips or dental floss impregnated with CHX [19,22]. Periodontal chips have shown promising results, suggesting that in conjunction with mechanical debridement, it can lead to clinical improvements in the signs and symptoms of peri-implant mucositis (reduction in the number of bleeding sites around implants and probing pocket depths) [23,24]. CHX is also used in endodontic treatment, but at a higher concentration (2%) for prolonged antibacterial action and neutralization of the demineralization/decalcification effect of primary irrigates such as sodium hypochlorite; CHX has no tissue dissolution properties, and infiltrates dentine [25]. CHX is most commonly promoted as a mouthwash solution with concentrations ranging from 0.1% to 0.2% [16]. In the systematic review published by Berchier et al., it was mentioned that the same antiplaque efficacy can be obtained, regardless of the dose used, if adjustments are made in terms of the volume used, so that the amount of CHX is between 18–20 mg per use [26]. However, it should be taken into consideration that a higher amount of the substance may increase the incidence of adverse reactions. This has led to the production of mouthwashes with a concentration of 0.05% or 0.06% CHX, which can provide therapeutic effect with a minimum incidence of adverse reactions [13,16,26]. Studies have shown that clinically utilized concentrations of chlorhexidine (CHX) at 2.0% exhibit cytotoxic effects on osteoblasts, fibroblasts and myoblasts (in vitro study) [27]. Even at concentrations as low as 0.002%, CHX has demonstrated cytotoxicity and inhibited cell migration across all cell types, highlighting its profound cytotoxic potential at significantly lower concentrations than those used in clinical practice. While CHX remains an effective topical antiseptic agent when applied as directed prior to surgery, further in vivo studies are necessary to investigate its impact on wound and tissue healing when used in proximity to open incisions, applied on postoperative dressings, or directly within wounds [27].

Regarding its toxicity at higher dosages, chlorhexidine has been associated with some potential adverse effects. Prolonged or excessive use of chlorhexidine may lead to side effects such as staining of teeth, tongue and restorations, altered taste perception (metallic taste), oral mucosal irritations and, in rare cases, allergic reactions [28]. Additionally, some studies suggest the long-term use of chlorhexidine may disrupt the natural oral microbiota balance, potentially leading to an overgrowth of opportunistic pathogens [29].

Although some studies have shown that there are no differences in the microbial composition of bacterial plaque in dental implants compared to natural teeth [30], certain factors influence the deposition of bacterial plaque and the evolution of peri-implant inflammation. These factors include the material from which the implant is made, the abutment–implant interface, implant roughness, absence of defensive elements from the gingival fluid and desmodontium, and the presence of teeth with periodontal damage [4,30]. Studies also aim to compare the microbiologic profiles of peri-implantitis and periodontitis, despite the limitations caused by the heterogeneity of the existing studies [31,32].

Furthermore, these studies [31,32] suggest that peri-implantitis is characterized by the presence of aggressive and resistant microorganisms, which distinguishes it from periodontitis. However, due to the variations in study methodologies, a comprehensive comparison of the results is challenging. Thus, while acknowledging the limitations of existing studies, the findings highlight the significance of aggressive and resistant microorganisms in the microbiologic profile of peri-implantitis. Moreover, it emphasizes the importance of using quantitative characteristics of the microflora cohabitants as key determinants of the disease rather than the qualitative composition. To further advance our understanding of peri-implant diseases, future research should prioritize the adoption of new techniques to enhance microbial detection, ultimately leading to improved diagnostics and targeted interventions in the management of peri-implantitis.

In the specialized literature, there are data about the impact of CHX on periodontal parameters, but less information about when it is used in peri-implant conditions. In recent years, implant–prosthetic therapy has exponentially increased, requiring more detailed protocols for investigating the status of peri-implant soft tissue, as well as non-surgical therapeutic modalities such as the use of CHX mouthwashes in peri-implant inflammatory diseases [33].

Peri-implant mucositis occurs when bacterial biofilms build up around dental implants, resulting in inflammation. Given the absence of reliable treatments for peri-implantitis, the current focus is on preventing the transition from peri-implant mucositis to peri-implantitis. This involves treating peri-implant mucositis and implementing early treatment procedures and protocols [34].

Non-surgical treatment refers to periodontal debridement, which consists of removing plaque, calculus and bacterial toxins from the tooth surface and below the gum line (root or implant surface) using manual and/or mechanical specialized instruments. It can be performed by the full-mouth or quadrant-wise treatment methods. Subgingival mechanical instrumentation, according to the 2018 Periodontal New Classification [1], has been considered the standard treatment for periodontal diseases. However, variations of this procedure have emerged, including full-mouth scaling (FMS), full-mouth disinfection (FMD) and quadrant-wise debridement (Q-SRP) in the context of periodontal treatment.

FMS involves the removal of plaque and calculus from all tooth surfaces in a single session, usually performed under local anesthesia. It offers the advantage of providing complete and immediate removal of dental deposits throughout the mouth [35]. FMD is an antimicrobial approach that combines subgingival mechanical instrumentation with adjunctive systemic or local antimicrobial agents. It aims to eliminate pathogenic microorganisms and to reduce the bacterial load in periodontal pockets. However, the success of FMD is dependent on proper patient compliance and the selection of appropriate antimicrobial agents [36]. Q-SRP is an approach that divides the mouth into quadrants, treating one quadrant at a time during separate appointments. This method allows for a more manageable treatment session, reduces patient discomfort and facilitates optimal oral hygiene maintenance [37]. However, the time required to complete treatment and the potential for reinfection in untreated areas between appointments are notable drawbacks. Each of these periodontal treatment approaches has its advantages and considerations. FMS provides immediate treatment, while FMD incorporates antimicrobial agents to improve the oral microbiota. Q-SRP offers a more segmented and manageable treatment approach. Clinicians should consider factors such as patient compliance and individual patient needs when selecting the most appropriate treatment modality.

These treatment methods mentioned—full-mouth scaling (FMS), full-mouth disinfection (FMD) and quadrant-wise debridement (Q-SRP)—are primarily focused on the treatment of periodontitis cases. Peri-implant mucositis, on the other hand, specifically refers to inflammation and infection around dental implants.

While these treatment methods may have some applicability to peri-implant mucositis, it is important to note that the management of peri-implant diseases requires specific considerations. The treatment of peri-implant mucositis typically involves the removal of bacterial biofilms and the reduction of inflammation in the soft tissues surrounding the implant. However, the methods and protocols used are comparable to the techniques utilized in the treatment of periodontal diseases.

Treatment approaches for peri-implant mucositis often involve mechanical debridement, such as the use of plastic or metal curettes, ultrasonic scalers or air-abrasive devices. These instruments are used to carefully remove biofilms and plaque from the implant surface and the surrounding mucosa. The air abrasive device with low-abrasive air-polishing erythritol powder (Perio Plus) is designed for peri-implant mucositis periodontal debridement [38]. The low granulation powder ensures the gentle effective removal of biofilm and debris from the implant surfaces and surrounding tissues. This targeted approach helps promote healing, reduces inflammation, and maintains the overall health of the peri-implant area.

In some cases, local antimicrobial agents or antiseptics may be utilized to aid in the disinfection process. Additionally, oral hygiene instructions and supportive care play a crucial role in the long-term management of peri-implant mucositis.

It is essential to consult with a dental professional who specializes in periodontics and implantology to determine the most appropriate treatment approach for peri-implant mucositis. In summary, while FMS, FMD and Q-SRP are primarily associated with the treatment of periodontal diseases, certain aspects of these methods may be relevant to the management of peri-implant mucositis. However, the treatment of peri-implant mucositis requires a specific approach that focuses on the characteristics and challenges associated with dental implants.

## 2. Materials and Methods

A research survey within the PubMed and Scopus databases from 2016 to 2021 was performed. The keywords followed were chlorhexidine OR chlorhexidine di-gluconate OR chlorhexidine gluconate OR CHX OR CHX AND mouthwash OR mouthrinse AND peri-implant mucositis OR dental implant OR mucositis OR anti-microbial OR non-surgical treatment OR anti-infective therapy.

The focused question was performed based on the PICOS format: Does the utilization of various treatment protocols involving chlorhexidine influence the microbial plaque formation and clinical outcomes in adult patients with peri-implant mucositis? For the selection of scientific articles relevant to the research topic, the PICOS criteria (population, intervention, comparison, outcome and study design) were followed. There were randomized controlled trials included (study design) involving adult subjects (>18 years) diagnosed with peri-implant mucositis (population); these were divided into a test group in which CHX was used in combination with debridement (intervention); there was at least one control group in which debridement was performed without the association of CHX (comparison); and for these groups, a series of periodontal parameters were evaluated (outcome). In vitro studies, laboratory animal studies, retrospective studies, case series, non-English articles, non-debridement studies and non-surgical adjuvant therapies such as laser therapy were excluded, also ozone therapy, photodynamic therapy or those operated by surgical techniques.

Following the analysis of the titles, abstracts and protocols of each study, only 16 articles [3,31,32,33,34,35,36,37,38,39,40,41,42] were selected for detailed evaluation.

## 3. Results

After the evaluation of the 16 articles in their entirety, 9 studies were excluded.

The reasons for study exclusions were pilot studies [43,44,45]; studies involving the use of CHX not in combination with non-surgical periodontal therapy [46]; CHX also being administered to the control group as well as the test group (CHX) [3]; a CHX concentration used that was significantly higher (20%) than that used in the rest of the studies [23,24,42]; use of CHX in combination with another antiseptic substance (may influence the study results) [45,47]; cases of peri-implantitis being evaluated [42]; and use of only CHX in combination with non-surgical periodontal therapy [43]. Thus, seven studies were included in this analysis [39,40,41,42,45].

All patients, regardless of the group of interest, were diagnosed with peri-implant mucositis and received non-surgical treatment. Patients in the study population group were given CHX in the form of a mouthwash or gel for oral rinsing, used in the oral irrigator or for brushing instead of toothpaste [23,42,45,47,48]. The biggest difference in terms of study protocols was the follow-up period which ranged from 3 months to 12 months [45,46,49,50]. The main features of the studies accepted in this analysis are presented in Table 1.

### 3.1. Non-Surgical Treatment

Considering the analyzed studies, it was observed that full-mouth debridement treatment was performed in four studies [41,42,49,50]. Also, only three studies exclusively used manual instrumentation [40,41,48] while in the others, ultrasonic instrumentation was used [42,49] or manual instrumentation was associated with ultrasonic instrumentation [45]. Furthermore, there was insufficient evidence to support the effectiveness of one treatment method compared to others, as all of them were performed correctly.

### 3.2. Information Extraction

Data extraction began with the verification of titles and summaries (abstracts) of the selected articles followed by a full-text analysis to decide whether the selected manuscripts accomplished the inclusion criteria for the study.

### 3.3. Interpretation of Peri-Implant Parameters

The selected studies used the criteria included in Figure 1 to assess the treatment of mucositis.

### 3.4. Plaque Index

The analysis of all seven studies showed that the use of CHX significantly reduced the accumulation of bacterial plaque compared to the control groups in which other active substances or a placebo were used [39,40,41,42,48]. This was possible even if different concentrations of CHX were used, from 0.03% to 0.2%. Reductions in biofilm accumulation, expressed by plaque index, were identified in the test groups, with values between 7% and 20% at 3–4 weeks after use; subsequently, in the interval of 3–12 months, a slight increase was registered in all cases, from 20% to 29% [39,41,42,48]. In each study, a certain concentration of CHX versus placebo or another antiplaque agent was investigated, but different concentrations of CHX were not analyzed [39,45,46,47,48,49,50].

There are few studies that followed the results of using different concentrations of CHX; one of them is that of Haydari et al., who in 2017 conducted a randomized double-blind clinical study on 60 subjects with experimentally induced gingivitis, trying to highlight the impact of concentrations of CHX from commercial mouthwashes on plaque index, bleeding and side effects [13]; the CHX concentrations subjected to comparison were 0.2%, 0.12% and 0.06%, respectively, in solution without alcohol [13]. The study included subjects without known general conditions, of both sexes aged over 18 years, who had in each of the first and second quadrants at least 3 teeth without periodontal damage to the canine, first premolar, second premolar and first molar [13]. At the same time, there were excluded from the study the following: smokers, pregnant or breastfeeding women, those with known chronic diseases, with acute diseases in the oral cavity or those who followed local or general drug treatments in the last 3 months [13]. Before the beginning of the study period, all of the participants were professionally cleaned using silicone cups, were offered the same oral care kit, and used plastic tooth guards adapted to the dental crowns in quadrant 1 for application before brushing with instructions for use [13]. The recordings were made at 7, 14 and 21 days when the study was completed [13].

The results showed statistically significant differences only in the values of the Löe and Silness plaque index when using mouthwash with 0.2% CHX concentration compared to a concentration of 0.12% or 0.06%, but no difference was found between the formulas with concentrations of 0.12% and 0.06% [13]. Regarding the Löe and Silness gingival index, no significant differences were recorded among the three concentrations used [13]; however, there were differences in the occurrence of adverse reactions, with statistical significance in the case of changes in taste and numbness when using the 0.2% concentration formula compared to the other two types of mouthwash [13].

Therefore, the results obtained in the seven studies included in the analysis can be attributed to the fact that all subjects received indications for oral care and moreover, they underwent initial periodontal therapy, including the debridement phase (stage) [39,45,46,47,48,49,50].

### 3.5. Bleeding on Probing

In all seven studies, a reduction in bleeding on peri-implant probing was found [33,34,35,36,37,41]. These studies reported varying degrees of bleeding at different time points ranging between 9% and 48% at 3–12 weeks after the beginning of the study, subsequently recording values between 9% and 76% 3–6 months after the first visit [39,42,48,50,51,52]. During the observation period of the subjects, Menezes K. et al. found a slight increase in the bleeding on probing index at 6 months compared to the control at 3 months, with the index increasing from 40% to 45% in one study [39] and from 48% to 76% in another study (8). In all of the other studies, a decrease in the bleeding index was maintained throughout the observation period [40,41,42,48].

However, the question is whether this decrease was strictly due to the use of CHX or to the entire protocol treatment, as none of the other studies identified different statistically significant differences in bleeding on probing between test and control groups [39,40,41,42,48].

These conclusions were also reached by other researchers. Porras et al., in subjects with peri-implant mucositis, found a significant reduction in bleeding on probing in both the test group and the control group [51]. The same correlation was found in the study of Machtei et al., which was performed on patients with peri-implantitis [52]. However, there were also studies performed on subjects with mucositis that showed statistically significant differences in bleeding on probing between the test and control groups. For example, Thone-Muhling et al. found a much greater reduction in bleeding on probing in patients who used CHX compared to those who received a placebo over an 8 month observation period [53]. The study by Crespi et al., which evaluated the bleeding on probing in peri-implantitis patients with topical administration of CHX associated with mechanical therapy versus placebo, indicated a statistically much greater reduction in bleeding on probing when using CHX versus placebo [54]. This may also suggest a correlation between the effects of CHX and the degree of peri-implant tissue damage.

### 3.6. Probing Depth

All of the studies [40,41,42,48,50] also assessed the depth of the peri-implant probing, but no major benefit was identified in this regard with the use of CHX compared to placebo or other active substances [39,40,41,42,48]. This can also be attributed to the cases studied, as the subjects diagnosed with peri-implant mucositis did not have probing depths greater than 5 mm, and benefited from the initial therapy and instructions for oral hygiene. At the same time, these data are in accordance with the current specialized literature included in the systematic review conducted by Siyan Liu et al. and the one conducted by Alex Solderer et al. [14,28].

### 3.7. Subgingival Microbiome

The main objective for one of the studies was to evaluate the subgingival microbiome in patients with peri-implant mucositis [9]. At baseline, the peri-implant subgingival plaque showed a higher proportion of *Haemophilus* and *Neisseria* genera [9]. Significant microbial changes were found in peri-implant subgingival plaque after using delmopinol and CHX in association with peri-implant debridement [9]. After 1 month, the proportion of *Streptococcus* genus was significantly higher in the CHX and delmopinol groups [9]. The changes were significant at the first follow-up (1 month), but at the second follow-up and at 3 months, only the CHX group maintained a significant decrease in bacterial species richness [9].

## 4. Conclusions

The present study aimed to highlight the influence of CHX on the response to non-surgical therapy in patients with peri-implant mucositis. Despite the limitations, regarding the different follow-up periods, the limited number of participants and using different concentrations of CHX, the studies showed a positive influence on the reduction in bacterial plaque accumulation and on the reduction in mucosal inflammation, manifested by reduced bleeding on probing, although a complete remission of inflammation was not achieved in all cases.

No correlation was found between the use of CHX and a potential reduction in probing depth of the peri-implant mucosa, and the results did not show statistically significantly differences compared to other antiseptics. Further studies need to be conducted to evaluate different concentrations of CHX used under similar conditions or in combination with other substances. This analysis will aim to determine whether an increase in concentration will have a better impact on the condition of peri-implant tissues. The major limitations observed in the included studies are differences in study designs and methodologies; variations in the treatment protocols used; different concentrations and forms of chlorhexidine; limited sample sizes (control and study groups); short follow-up periods; and a lack of standardized outcome measures and criteria. To overcome the limitations observed, future studies should consider the following aspects: conducting well-designed studies with larger populations and longer follow-up periods; and standardizing treatment protocols, including the concentration, form and frequency of chlorhexidine application.

The future of scientific research regarding the adjuvant use of chlorhexidine in the non-surgical treatment of peri-implant mucositis could have immense potential. Several investigations could be pursued to enhance the understanding of its proper therapeutic approach. In the first place, rigorous clinical studies must be conducted to assess the efficacy and safety of chlorhexidine in treating peri-implant mucositis. These studies could involve long-term follow-ups to evaluate the sustained effects of the treatment. Additionally, exploring the antimicrobial properties of chlorhexidine could lead to an understanding of its mechanism of action, and optimize its use in preventing biofilm formation around implants. Also, further investigations could focus on elucidating the precise dosage, treatment protocols, and duration necessary to achieve optimal outcomes. Moreover, understanding the impact of chlorhexidine on wound healing, soft tissue regeneration and host response could contribute to developing therapeutic approaches. Innovative techniques such as in vitro studies, genetic analysis and microbiome profiling could provide insights into the microbial diversity and resistance patterns associated with peri-implant mucositis, and guide the development of personalized treatment strategies.

## Figures and Tables

**Figure 1 healthcare-11-01918-f001:**
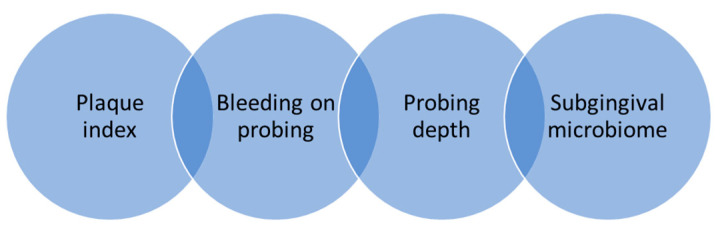
The evaluation criteria for the effectiveness of CHX in mucositis treatment.

**Table 1 healthcare-11-01918-t001:** Studies accepted in the analysis.

Authors	Year of Publication	Type of Study	No Subjects/No Sites with Mucositis	Protocol	Results	Follow-Up
Menezes et al. [39]	2016	RCT	37/61 test group, 58 control group	Instructions for oral care, removal of local retentive factors, supra- and subgingival debridement, irrigation test group CHX 0.12% twice daily, 15 mL, 30 min after brushing or placebo in the control group, 14 days.	Statistically significant decrease in PD, BOP, GBI and PI in both groups. CHX did not influence the results.	1, 3, 6 months
Hallstrom et al. [48]	2017	RCT	38/19 for each group	Debridement, oral care instructions, with toothpaste 0.2% NaF, to which was added CHX 0.2% to the test group, for 12 weeks.	At the beginning of the study, no significant differences. At 12 weeks, significantly greater reductions in PI, BOP and PD were observed in the test group.	4 and 12 weeks
Bunk et al. [40]	2020	RCT	60/20 for each group	Supra-/upper and subgingival debridement and instructions for oral care, group 1—oral rinses with placebo, groups 2 and 3—rinses by waterpik with water, respectively, CHX 0.06% 1/day, 12 weeks.	Complete resolution of inflammatory signs in 70% of cases at 12 weeks. Significantly better results in case of CHX combination. Mucositis prevalence at the end of the study—group 1: 50%, group 2: 25%, group 3: 5%	4, 8 and 12 weeks
Alzoman et al. [41]	2020	RCT	48/16 for each group	Instructions for oral care, debridement, group 1—oral rinses with water, group 2—oral rinses with herbal mouthwash, group 3, oral rinses with CHX 0.12%, twice daily, 14 days.	At group 1 without improvements compared to baseline, in groups 2 and 3 the PI, BOP and PD decreased, at 3, 6 and 12 weeks, but without differences between the 2 groups.	3, 6, 12 weeks
Philip et al. [42]	2020	RCT	89/31 group 1, 30 group 2, 28 group 3	Supragingival scaling, professional brushing, debridement of peri-implantation sites, group 1—oral rinses with 0.2% delmopinol hydrochloride, group 2—0.2% CHX, group 3—placebo	At 3 months, statistically significant decrease in BOP I%, mBI, no differences between the 3 groups.	1 and 3 months
De Melo Menezes et al. [49]	2021	RCT	30/47 group 1 and 49 group 2	Basic periodontal therapy consisted of individual guidance of oral hygiene, debridement of peri-implant sites and teeth. After instrumentation, group 1—oral rinses with CHX 0.12%, group 2—placebo, twice daily, 14 days	No significant statistical differences between the test and control group regarding VPI, GBI, BOP, PD or KMW. Significantly lower BOP at 3 months in the test and control groups with an increase at 6 months.	1, 3 and 6 months
Philip et al. [50]	2021	RCT	89/89 at baseline	Peri-implant debridement associated with oral rinses for 1 month as follows: group 1—delmopinol hydrochloride, group 2—CHX0.2%, and group 3—placebo. After 1 month, supragingival maintenance care was provided as well as after 3 month follow-up.	The clinical outcomes are summarized in the previous study (Philip et al. 2020). At 3 months, all 3 groups showed significant improvements in the clinical parameters (PI, BI, BOP and PD). Significant microbial changes were found in peri-implant subgingival plaque after using delmopinol and CHX. The changes were significant at first follow-up, but at second follow-up only the CHX group still maintained significantly lower bacterial species richness.	1 and 3 months

RCT—randomized controlled trial; CHX—chlorhexidine; PI—plaque index; BOP—bleeding at probation; PD—depth probing; GBI—gingival bleeding index; IBOP—bleeding at peri-implant probing; mBI—modified bleeding index; VPI—visible plaque index; KMW—keratinized mucosa width.

## Data Availability

Data are contained within the article.

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
