# Peer review of "The Effectiveness of the Association of Chlorhexidine with Mechanical Treatment of Peri-Implant Mucositis"

_healthcare, 2023, doi:10.3390/healthcare11131918_

Round 1

Reviewer 1 Report

Comments are in the pdf

Comments are in the pdf

Author Response

Thank you for your observation. I have attached a document with answers to your questions. All the aspects related to the study have been added in the revised version of the manuscript. 

Reviewer 2 Report

  In my opinion the work needs major reviewing before publication.

The PICOS strategy used is missing the PICOS question.

The fact that the mechanical treatment of mucositis had different   protocols, in  the included studies (curettes, air powder), is a major limitation to determine the real impact of CHX.

I think that some theoretical concepts should be reviewed and updated because they do not correspond to the latest state of knowledge:

Line 40- “The most suggestive clinical sign for the diagnosis of peri-implant mucositis is bleeding on probing.”

According to the World Workshop on the Classification of Periodontal and Peri-Implant Diseases and Conditions (2018), peri-implant mucositis can be diagnosed based on the following criteria: (1) presence of peri-implant signs of inflammation (redness, swelling, line or bleeding within 30 second after probing), combined with (2) no additional bone loss following initial healing.

Line 82- “Although studies have shown that there are no differences in the microbial composition of bacterial plaque in dental implants compared to natural teeth.”

3 systematic reviews report the microbiological findings of peri implantitis and conclude that the microbiological profile of peri implantitis differs from periodontitis being more complex and variable. It consists of resistant and aggressive mycoorganisms that may include opportunistic bacteria, fungi and viruses.

Padial-Molina et al Microbial Profile and Detection Techniques in Peri-Implant Diseases: a Systematic Review. J Oral Maxillofac Res. 2016;7:e10.

 Perez-Chaparro PJ  et al. The Current Weight of Evidence of the Microbiologic Profile Associated With Peri-Implantitis: A Systematic Review. J Periodontol 2016;87:1295-304.

 Rakic M et al The Microbiologic Profile Associated with Peri-Implantitis in Humans: A Systematic Review. Int J Oral Maxillofac Implants. 2016;31:359-68. Peri-implant diseases

Line 102- “Scaling and root planing (SRP) have been considered the standard treatment for periodontal diseases.”

According to the World Workshop on the Classification of Periodontal and Peri-Implant Diseases and Conditions (2018), the term SRP was substituted by Subgingival Mechanical Instrumentation.

In my opinion, the comparison made between treatment of periodontitis and peri-implantitis, namely Mucositis, does not make much sense in terms of full mouth disinfection, in patients with unitary or partially edentulous patients, with gingivitis and mucositis.

I also advice the authors to include in the table of results, of the included studies, the number of implants diagnosed with Mucositis and interventioned.

Line 145-“A research within PubMed and Scopus databases from the last five years (2016 - 2021) was performed.”

 The search period is outdated, the expression last 5 years should be removed.

Line 180-“All patients, regardless of the interested group, were diagnosed with peri-implant mucositis and received non-surgical treatment. Patients in the target groups were given CHX in the form of mouthwash or gel, for oral rinsing, used in the oral irrigator or for brushing instead of toothpaste.”

Study 49 - Efficacy of a 0.03% chlorhexidine and 0.05% cetylpyridinium chloride mouth rinse in reducing inflammation around the teeth and implants: a randomized clinical trial

I believe this study does not meet the eligibility criteria because the mouth rinse tested, in addition to 0.03% CHX has 0.05% cetylpyridinium chloride in its composition.

 About this theme I also advise the authors to read and include the following article: Liu JX, Werner J, Kirsch T, Zuckerman JD, Virk MS. Cytotoxicity evaluation of chlorhexidine gluconate on human fibroblasts, myoblasts, and osteoblasts. J Bone Jt Infect 2018;3(4):165–172

“CHX has specific disadvantages because clinically used 2% CHX permanently halts cell migration and signifi­cantly reduces fibroblast, myoblast, and osteoblast survival in vitro. Thus, further in vivo studies are required to exam­ine and optimize CHX safety and efficacy.”

The present article, in my opinion, needs a major revision in  English language. In the course of the text are found outdated words and expressions with little scientific rigor. Some examples:

Line 35- “the treatment of edentulous through”

Line 42- “bacterial development,”

Line 51- “At present, the idea accepted is that a correct brushing technique is sufficient to keep plaque and gingival bleeding indices within normal limits, even in the absence of

using the mouthwash with anti-plaque active compounds”

Line 88- “In the literature, there are many dates”

Line 181- “target groups”

Author Response

(The authors gave the same response as above.)

Reviewer 3 Report

Chlorohexidine has been considered as gold standard for its ideal use as prophylactic antimicrobial agent. Lot of work has been done. The review needs still to cover a lot many aspects to make it complete.

1. The most recent intervention is this area is discovery and use of chlorohexidine chip therapy

2. More elaborations is expected in the area of 

3. Dosage recommendation,

4. toxicity at higher dose and prolonged use

Needs Improvement is English gramer and language

Author Response

(The authors gave the same response as above.)

Round 2

Reviewer 2 Report

I am satisfied with the changes made by the authors contributing to a more updated, rigorous, and improved manuscript version with correct written  english.

Reviewer 3 Report

The review covers recent approach to old science and its state of the art.